# Design, Synthesis, Antibacterial, Antifungal and Anticancer Evaluations of Novel *β*-Pinene Quaternary Ammonium Salts

**DOI:** 10.3390/ijms222011299

**Published:** 2021-10-19

**Authors:** Li Zhang, Xue-Zhen Feng, Zhuan-Quan Xiao, Guo-Rong Fan, Shang-Xing Chen, Sheng-Liang Liao, Hai Luo, Zong-De Wang

**Affiliations:** 1East China Woody Fragrance and Flavor Engineering Research Center of National Forestry and Grassland Administration, Camphor Engineering Research Center of National Forestry and Grassland Administration, College of Forestry, Jiangxi Agricultural University, Nanchang 330045, China; zhangli_miaomuli@163.com (L.Z.); fgr1971@163.com (G.-R.F.); csxing@126.com (S.-X.C.); liaosl@jxau.edu.cn (S.-L.L.); 2National Engineering Laboratory for Biomass Chemical Utilization, Institute of Chemical Industry of Forest Products, Chinese Academy of Forestry, Nanjing 210042, China; xuezhen__feng@163.com; 3College of Chemistry, Jiangxi Normal University, Nanchang 330022, China; xiaozhq3501@163.com

**Keywords:** *β*-pinene, quaternary ammonium salts, antifungal activity, antimicrobial, anticancer, preliminary antimicrobial mechanistic

## Abstract

*β*-pinene is a monoterpene isolated from turpentine oil and numerous other plants’ essential oils, which has a broad spectrum of biological activities. In the current work, six novel *β*-pinene quaternary ammonium (*β*-PQA) salts were synthesized and evaluated in vitro for their antifungal, antibacterial and anticancer activities. The in vitro assay results revealed that compounds **4a** and **4b** presented remarkable antimicrobial activity against the tested fungi and bacteria. In particular, compound **4a** showed excellent activities against *F. oxysporum f.sp. niveum*, *P. nicotianae var.nicotianae*, *R. solani*, *D. pinea* and *Fusicoccumaesculi,* with EC_50_ values of 4.50, 10.92, 9.45, 10.82 and 6.34 μg/mL, respectively. Moreover, compound **4a** showed the best antibacterial action against *E. coli*, *P. aeruginosa*, *S. aureus* and *B. subtilis*, with MIC at 2.5, 0.625, 1.25 and 1.25 μg/mL, respectively. The anticancer activity results demonstrated that compounds **4a**, **4b**, **4c** and **4f** exhibited remarkable activity against HCT-116 and MCF-7 cell lines, with IC_50_ values ranged from 1.10 to 25.54 μM. Notably, the compound **4c** displayed the strongest cytotoxicity against HCT-116 and MCF-7 cell lines, with the IC_50_ values of 1.10 and 2.46 μM, respectively. Furthermore, preliminary antimicrobial mechanistic studies revealed that compound **4a** might cause mycelium abnormalities of microbial, cell membrane permeability changes and inhibition of the activity of ATP. Altogether, these findings open interesting perspectives to the application of *β*-PQA salts as a novel leading structure for the development of effective antimicrobial and anticancer agents.

## 1. Introduction

Microbial (such as bacteria, fungi and viruses) infection is a major threat to human health and also the primary cause of both plant diseases and losses of crop production post-harvest [1,2,3]. To guard against microbial pathogens, the chemical fungicides are still the primary approach, which are economical, efficient and have played an indispensable role in nourishing more people throughout human history [4]. Unfortunately, the widespread and injudicious use of antimicrobial agents has resulted in the emergence of agent resistances, environmental hazards and many other drawbacks [5]. Thus, novel antifungal agents that are ecologically friendly and have high efficiency are needed. Simultaneously, the increasing demand of novel, environmental-friendly and low-toxicity antimicrobial agents have driven many studies about the botanical fungicides. Lots of research results showed that botanical fungicides were a potential replacement for pure chemical synthetic pesticides in agriculture and organic food production, with good biocompatibility, rapid biodegradation speed, less or no cross-resistance and more diverse structures [6,7]. There are many bioactive secondary metabolites in plants, and the essential oil is considered to be one of the most bioactive secondary metabolites among them [8,9]. The essential oil of plants has various biological effects and also been proven to play an important role in microbial action [10,11].

*β*-pinene is one of the chief constituents of turpentine oil and numerous of other plants’ essential oils, belonging to the best-known representatives of a broad family of monoterpenes [12]. A wide range of pharmacological activities of the *β*-pinene have been reported, including antibiotic resistance modulation, anticoagulant, antimicrobial, antimalarial, antitumor and anti-inflammatory activities [13]. Quaternary ammonium compounds (Quats) are a large class of permanently charged cationic chemicals [14]. Quats have been reported to possess a large variety of biological activities, including antibacterial, antifungal, anti-parasite and antivirus activities [15,16]. Moreover, the quaternary ammonium scaffolds are crucial to antimicrobial agents and have been used in a variety of consumer and industrial products for their antimicrobial properties, such as benzalkonium chloride, dequalinium chloride and cetylpyridinium chloride [17,18,19].

Herein, based on the previous research foundation, six *β*-PQA salts were synthesized and their antifungal, antibacterial and anticancer activities were also evaluated. Moreover, the possible preliminary antimicrobial mechanism was further evaluated by scanning electron microscopy observations, the relative electric conductivity experiments and the ATP activity inhibition experiments. This exploration is expected to improve the added value of *β*-pinene and its derivatives as antimicrobial and anticancer agents.

## 2. Results

### 2.1. Chemistry

According to our previous work, six *β*-PQA salts were successfully prepared by using *β*-pinene as a precursor with an efficient and flexible method under mild conditions that obtained the desired products with high yields (Figure 1). The structures of target compounds **4a**–**4f** were confirmed by ^1^H NMR and ^13^C NMR, FTIR and HRMS (see the Experimental Section and Appendix A). In the ^1^H NMR spectra of all compounds, the proton signal of the CH_2_ or CH_3_ group attached to the N group was observed at δ 3.381–3.975 ppm. For compounds **4d**–4f, the ^1^H NMR spectrum showed a peak at 3.87 ppm related to O-CH_2_. In the ^13^C NMR spectra of all compounds, the carbon signal of the CH_2_ or CH_3_ group attached to the N group was observed at δ 51.19–62.82 ppm. As for the compound **4a**, this was seen at δ 51.79 and 62.82 ppm. The ^13^C NMR spectra of compounds **4d**–4f exhibited peaks at δ 60.09–63.10 related to the C-O bond. The FTIR spectra for compounds **4a**–**4f** showed peaks nearby at 2997–2867, 1450 and 1227 cm^−1^ related to *V* C-H, *δ* C-H and C-N, respectively. The HRMS spectrum of **4a**–**4f** showed the expected fragments’ ion peaks at m/z 346.34 (M^+^-Cl), 346.34 (M^+^-I), 386.37 (M^+^-I), 388.35 (M^+^-Cl), 388.35 (M^+^-Br) and 388.35 (M^+^-I) respectively, which was the same as the predicted molecular weight.

### 2.2. Antifungal Activity

In this study, the in vitro antifungal activities of the target compounds were assayed against *F. oxysporum f.sp. niveum*, *P. nicotianae var.nicotianae*, *C. acutatum*, *R. solani*, *C. versicolor*, *F. verticillioides*, *D. pinea*, *P. vaporaria*, *Fusicoccumaesculi* and *C. gloeosprioides*. Chlorothalonil were selected as the reference drugs. Most of the *β*-PQA salts showed moderate to excellent in vitro activities (Table 1 and Appendix A Appendix A). The inhibition rate of the tested fungi increased with the increasing concentration of the compounds. At the concentration of 50 μg/mL, compounds **4a**, **4b** and **4c** could completely inhibit the growth of specific fungi, such as compound **4a** against *D. pinea*, compound **4b** against *P. vaporaria* and compound **4c** against *P. nicotianae var.nicotianae* and *P. vaporaria*. In addition, compounds **4a**, **4b** and **4c** exhibited broad-spectrum antifungal activity with the EC_50_ values of 4.50–33.76, 5.39–27.83 and 9.36–77.68 μg/mL, respectively. It was notable that compound **4a** showed excellent antifungal activities against *F. oxysporum f.sp. niveum*, *P. nicotianae var.nicotianae*, *R. solani*, *D. pinea* and *Fusicoccumaesculi,* with EC_50_ values of 4.50, 10.92, 9.45, 10.82 and 6.34 μg/mL, respectively (Figure 2 and Table 1). Significantly, compound **4a** had a 60.09% inhibition rate against *F. oxysporum f.sp. niveum* at the concentration of 6.25 μg/mL, with the EC_50_ value of 4.50 μg/mL.

Subsequently, the preliminarily structure–activity relationships (SAR) of target compounds were analyzed. The changes of the different groups (two methyl groups, piperidine group and morpholine group) introduced on the hydrophilic group N^+^ of the *β*-PQA salts had different effects on the antifungal activity and antifungal spectrum. It was worth noting that introduction of two methyl groups (compounds **4a** and **4b**) or the piperidine group (compound **4c**) could significantly improve the antifungal activity compared to those compounds with the morpholine group (compounds **4d**–**4f**). In addition, compounds **4e** and **4d** with the halide ions (Br^−^ > Cl^−^) resulted in an increased antifungal activity compared to compound **4f** with halide ions (I^−^).

### 2.3. Antibacterial Activity

The *β*-PQA salts were also tested for their antibacterial activity against two Gram-positive (*S. aureus*, *B. subtilis*) and two Gram-negative (*E. coli*, *P. aeruginosa*) bacteria. From the results of bacterial growth inhibitory activity on the test strains (Table 2), it could obviously be found that most target compounds showed moderate to excellent antibacterial activity. The antibacterial potency could be presented in the following order: **4a** > **4b** > **4c** > **4e** > **4d** > **4f**. Compounds **4a** and **4b** showed strong activity on the four bacteria, with MIC values ranging from 0.625 to 80 μg/mL. Especially, compound **4a** showed the best antibacterial action against *E. coli*, *P. aeruginosa*, *S. aureus* and *B. subtilis,* with minimal inhibitory concentrations (MIC) at 2.5, 0.625, 1.25 and 1.25 μg/mL, respectively. Moreover, the antibacterial activity of compound **4a** against *E. coli* was superior to the positive control chloramphenicol (MIC: 5 μg/mL), and the antibacterial activity against *S. aureus* and *B. subtilis* was close to the positive control chloramphenicol (MIC: 0.625 and 0.625 μg/mL). 

The SAR of target compounds against antibacterial activity were similar to antifungal activity. The introduction of halide ions (Br^−^ > Cl^−^) and groups (two methyl groups, piperidine group) into the hydrophilic group N^+^ of the *β*-PQA salts was conducive to improve the antibacterial activity.

### 2.4. Anticancer Activity

Human health in the current medical era is facing numerous challenges, especially due to cancer. In this study, we also evaluated the anticancer activity of all the *β*-PQA salts by the CCK-8 method against HCT-116 and MCF-7 cell lines, using sorafenib as a positive control. The results were expressed as the IC_50_ (50% inhibitory concentration) values in μΜ (Table 3). It was found that the *β*-PQA salts demonstrated different levels of anticancer activity against different cell lines, and the anticancer effects of the *β*-PQA salts were dose-dependent (Figure 3 and Figure 4). In general, almost all the target compounds exhibited potent anticancer activity. It was clearly shown that as the concentration of the tested sample increased, the relative cell viability of cancer cells decreased (Figure 3 and Figure 4). As the HCT-116 and MCF-7 cell lines were treated with compounds **4a**, **4b**, **4c** and **4f**, all the compounds exhibited excellent anticancer activity, with IC_50_ values ranging from 1.10 to 25.54 μM (Table 3). The anticancer activity of compounds **4a**, **4b** and **4c** were superior to the positive control sorafenib (Table 3). In particularly, the compound **4c** displayed the strongest cytotoxicity compared to other compounds, with the IC_50_ values of 1.10 and 2.43 μM against HCT-116 (Figure 3C) and MCF-7 (Figure 4C) cell lines respectively, which were 18.90- and 5.73-folds higher than sorafenib (IC_50_ = 20.79 and 13.92 μM). 

SAR analysis of these compounds revealed that the introduction of two methyl groups (compounds **4a** and **4b**) or the piperidine group (compound **4c**) on the hydrophilic group N^+^ of the *β*-PQA salts resulted in a significant increase of the anticancer activity as compared to compounds **4d**–4f with the morpholine group. However, the difference from the SAR of antifungal and antibacterial activity was that the compounds **4f** and **4e** with halide ions (I^−^ > Br^−^) showed better activity than compound **4d** with halide ions (Cl^−^). 

Taken together, the current report presents that compounds **4a**, **4b** and **4c** could be considered as promising broad-spectrum potent anticancer candidates for further development for new anticancer drugs, and the safety and anticancer target need to be further investigated.

### 2.5. Morphology Analysis

Based on the above research results, we found that compound **4a** had excellent antifungal and antibacterial activity, to elucidate the effective pathway of compound **4a** at the EC_50_ value of 4.5 μg/mL or MIC of 0.625 μg/mL against *F. oxysporum f.sp. niveum* and *P. aeruginosa*, the morphological changes of mycelium and strains were observed by SEM. As shown in Figure 5, the mycelium (Figure 5A) and strain (Figure 5C) were homogeneous, and the endosome of the cells appeared orderly arranged with a complete shape and a smooth surface in the control group. In contrast, after treating with compound **4a**, drastic changes in mycelium (Figure 5B) and strains’ (Figure 5D) morphology were observed: the cells were abnormal, the arrangement of the endosome appeared distorted and the surface shrunk and became rough with lots of holes. Therefore, the morphological changes of the mycelia and strains could be one of the antimicrobial mechanisms of compound **4a** on *F. oxysporum f.sp. niveum* and *P. aeruginosa*.

### 2.6. Relative Electric Conductivity

To further verify the conclusion of SEM about cell membrane integrity, the cell membrane permeability of *F. oxysporum f.sp. niveum* and *P. aeruginosa* was determined by testing the changes in relative electric conductivity of mycelium and strains’ suspensions with or without compound **4a**. As shown in Figure 6, the conductivity rates of the mycelium and strains’ suspensions treated with compound **4a** were obviously raised, higher than the control group, and increased time-dependently. Therefore, it was reasoned that compound **4a** affected the cell membrane permeability of *F. oxysporum f.sp. niveum* and *P. aeruginosa*. From the above results, it was inferred that compound **4a** destroyed the integrity of the cell membrane, causing an increase of cell membrane permeability and leading to the death of mycelium and strains.

### 2.7. ATP Activity

For further investigation on the effect of compound **4a** on energy metabolism for *F. oxysporum f.sp. niveum* and *P. aeruginosa*, the detection of intracellular adenosine triphosphate (ATP) activity was conducted. Interestingly, compared with the control group, the intracellular ATP activity of the strains treated with compound **4a** was decreased (Figure 7). Especially for *P. aeruginosa*, the intracellular ATP activity decreased significantly (*p* < 0.01) after treating with compound **4a** (Figure 7B). Therefore, it could be speculated that the ATP activity of *F. oxysporum f.sp. niveum* and *P. aeruginosa* treated by compound **4a** were markedly inhibited. Considering all the above results, compound **4a** can not only destroy the integrity of the cell membrane, but also affect the energy metabolism of the microbial. This can partly explain the antimicrobial mechanism of compound **4a**. Further explorations on the structural modification of compound **4a** and its specific mechanisms of action and safety are needed.

## 3. Discussion

Emergence of antimicrobial agent resistances is a serious problem worldwide, and there is an increased demand for novel antimicrobial agents that are ecologically friendly and have high efficiency. It is widely known that essential oils of plant origin may act as promising antimicrobial agents [20]. *β*-pinene is an important compound derived from plant essential oils, and its derivatives have significant research and development values on account of their wide range of pharmacological activities [13,21,22]. In our previous research, a series of quaternary ammonium salts were synthesized from *β*-pinene and we obtained many derivatives with potent antimicrobial activity [23]. However, no detailed studies have been investigated on its application in anticancer activities and antimicrobial mechanism. In the current study, six *β*-PQA salts were rationally designed, and most *β*-PQA salts had high and broad-spectrum antifungal and antibacterial activities (Table 1 and Table 2). It was found that *β*-pinene derivatives exhibited distinct anticancer activity [13,24]. In our study, the anticancer experiments provided strong evidence for the anticancer activity of *β*-PQA salts against HCT-116 and MCF-7 cell lines (Table 3). 

The structural types of compounds are few, which limits the systematic structure–activity relationship study. However, from the antifungal and antibacterial results, we have summarized some interesting findings. In this study, we found that the introduction of two methyl groups or a piperidine group was conducive to improving the antifungal and antibacterial activity, compared to those compounds with the morpholine group. Zhong et al. [25] found that the molecular weight, steric hindrance and alkyl chain length of the compound quaternary ammonium salts affected the antimicrobial activity. In our previous report, the antifungal activities of *β*-PQA salts with two methyl groups linked on the hydrophilic group N^+^ were stronger than those compounds with the two ethyl groups [23]. We speculate that lower molecular weight and steric hindrance compounds are easier to absorb and diffuse through the cell membrane, which increases the chance of these compounds entering the bacteria or fungi to exert antibacterial or antifungal effects. In addition, the anions contained in the quaternary ammonium salt will also affect its antibacterial effect [26]. Chen et al. [6] found that the quaternary ammonium salt with an anion of bromide had more potent antimicrobial properties than those with chloride anions [27]. In our study, we also found that the antibacterial and antifungal activities of *β*-PQA salts with a bromide anion were better than those compounds with chloride and iodide anions.

In the current investigation, compound **4a** showed remarkable antimicrobial potential against *F. oxysporum f.sp. niveum* and *P. aeruginosa*. Therefore, compound **4a** was selected for the antimicrobial mechanism study.

A previous study had shown that the essential oils could destroy the integrity, permeability and fluidity of the microbial cell membrane [28]. The main function of the cell membrane is to control the transportation of nutrients and metabolites, and maintain the normal life activities of the cell [29]. When the structure of the cell membrane is destroyed, many internal organelles and other components will leak out, which will eventually inactivate the cell, and then lead to the death of the host. According to previous studies, the mode of action of cationic biocides is binding to the cytoplasmic membrane, disruption of the cytoplasmic membrane, release of cytoplasmic constituents, etc. [25]. It had been reported that the derivatives of *β*-pinene could change mycelial morphology and increase cell membrane permeability of *Botrytis cinerea* [30]. Thus, the *β*-PQA salts may achieve their antimicrobial activity by destroying the cell membrane of microbials. It could be observed from the SEM that the cell membrane integrity of *F. oxysporum f.sp. niveum* and *P. aeruginosa* treated with compound **4a** were destroyed, and the cell membrane permeability was also increased (Figure 5 and Figure 6).

The essential oils can also affect the normal operation of the energy metabolism pathway of microbials, thereby inhibiting microbial growth [31]. Previous studies by Han et al. demonstrated that limonene could hinder the ATP synthesis by inhibiting the activity of the respiratory complex and ATPase against *Listeria monocytogenes* [32]. ATP is the energy currency and plays central roles in many biological reactions, and the depletion of ATP from tissues and cells is thus a sensitive marker of impaired cellular function and viability [33,34]. Antoci et al. found that the quaternary ammonium salts had an excellent quasi-nonselective antifungal activity against the fungus *Candida albicans* and the best-fit in complex with ATP synthase [35]. Therefore, the impact of compound **4a** on the energy metabolism for *F. oxysporum f.sp. niveum* and *P. aeruginosa* was detected by detecting ATP activity. We found that the ATP activity of *F. oxysporum f.sp. niveum* and *P. aeruginosa* treated with compound **4a** decreased, indicating that compound **4a** could disrupt their energy metabolism. Therefore, the energy metabolism disruption of microbials is probably one of the main antimicrobial mechanisms of *β*-PQA salts, especially for compound **4a**.

## 4. Materials and Methods

### 4.1. Materials and Structural Characterization Techniques

All chemicals (analytical grade) used were obtained from Sigma Aldrich (St. Louis, MO, USA) and Sinopharm Chemical Reagent Co., Ltd (Shanghai, China). Melting points were determined in an open capillary using a 44X-6T Micro melting point apparatus (Shanghai, China) and were uncorrected. All the ^1^H and ^13^C-NMR spectra were measured on a Bruker DKX500 NMR spectrometer (Bruker, Karlsruhe, Germany) using CDCl_3_ as a solvent. Chemical shifts were reported in ppm (δ). IR spectra were recorded on a Nicolet IS10 FTIR spectrometer (Nicolet, Madison, WI, USA). HR-MS spectra were recorded on a SCIEX UHPLC30A-Trip TOFTM 5600 mass spectrometer (Concord, Ontario, Canada). 

The ten tested fungi: *Fusarium oxysporum f.sp. niveum* (*F. oxysporum f.sp. niveum*), *Phytophthora nicotianae var.nicotianae* (*P. nicotianae var.nicotianae*), *Colletotrichum acutatum* (*C. acutatum*), *Rhizoctonia solani* (*R. solani*), *Coriolus versicolor* (*C. versicolor*), *Fusarium verticillioides* (*F. verticillioides*), *Diplodia pinea* (*D. pinea*), *Poria vaporaria* (*P. vaporaria*), *Fusicoccumaesculi* and *Colletotrichum gloeosprioides* (*C. gloeosprioides*), were isolated from susceptible plants. The four tested bacteria were *Escherichia coli* ATCC 25922 (*E. coli*), *Pseudomonas aeruginosa* ATCC9027 (*P. aeruginosa*), *Staphylococcus aureus* ATCC 25923 (*S. aureus*) and *Bacillus subtilis* ATCC 23631 (*B. subtilis*). The ten tested fungi and four tested bacteria were provided by the plant pathology laboratory from the College of Agriculture, Jiangxi Agricultural University. After retrieval from the storage tube, the fungi were incubated in PDA (potato dextrose agar) at 25 °C for a week to obtain new mycelia for the antifungal assay, and the bacteria were incubated in LB (Luria Bertani) liquid medium at 37 °C for 24 h.

### 4.2. General Procedure for the Synthesis of β-PQA Salts

Intermediate (1R, 2R, 5R)-hydronopyl halide (**2a**–**2c**) and (1R, 2R, 5R)-hydronopyl ammonium halide (**3a**–**3c**) were synthesized from *β*-pinene according to the process which was previously reported [23,36,37]. Then, the bis-hydronopyl quaternary ammonium salts (**4a**–**4f**) were obtained by the reaction between (1R, 2R, 5R)-hydronopyl halide (**2a**–**2c**, 0.05 M) and (1R, 2R, 5R)-hydronopyl ammonium halide (**3a**–**3c**, 0.05 M) in 30 mL ethyl acetate at 60 °C. After 48~72 h of reaction, the ethyl acetate was vacuum removed. Under three light petroleum washes and recrystallization from ethyl acetate, a white solid bis-hydronopyl quaternary ammonium salt (**4a**–**4f**) was obtained. The synthetic routes and structures of compounds **4a**–**4f** are shown in Figure 1.

Spectroscopic analysis of compound **4a**: bis-hydronopyl dimethyl ammonium chloride, white solid, m.p. 222.1–227.5 °C; Yield, 80%. ^1^H-NMR (CDCl_3_, 400 MHz) δH: 3.438 (t, J = 4.8 Hz, 4H, 2 _11_-CH_2_), 3.416 (s, 6H, 2 α-CH_3_), 2.321 (m, 2H, 2 _2_-CH), 1.943~1.820 (m, 2H, 12 (_10_-CH_2_, _7_-CH, _1_-CH, _5_-CH, _3_-CH)), 1.709 (m, 4H, 2 _4_-CH_2_), 1.428 (m, 2H, 2 3-CH), 1.184 (s, 6H 2 _9_-CH_3_), 1.001 (s, 6H, 2 _8_-CH_3_), 0.866 (d, J = 9.8 Hz, 2H, 2 _7_-CH); ^13^C-NMR (CDCl_3_, 100 Hz) δc 62.11 (2C-_11_), 51.50 (2C-_α_), 46.27 (2C-_2_), 40.88 (2C-_5_), 38.43 (2C-_6_), 38.18 (2C-_1_), 33.20 (2C-_10_), 30.03 (2C-_7_), 27.78 (2C-_9_), 25.91 (2C-_4_), 23.18 (2C-_8_), 21.96 (2C-_3_); 1R, y_max_ (cm^−1^): 3429; IR (KBr) *v* (cm^−1^) 3446.78, 2981.91, 2935.72, 2899.66, 2866.20, 1469.88, 1366.31; LC-MS, C24H44NCl, *m/z* 346.34 (M^+^-Cl).

Spectroscopic analysis of compound **4b**: bis-hydronopyl dimethyl ammonium iodide, white solid, m.p. 224.5–230.8 °C; Yield, 82%. ^1^H-NMR (CDCl_3_, 400 MHz) δH: 3.407 (m, 4H, 2 _11_-CH_2_), 3.286 (s, 6H, 2 α-CH_3_), 2.243 (m, 2H, 2 _2_-CH), 1.959~1661 (m, 16H, 2 (_7_-CH, _10_-CH_2_, _5_-CH, _1_-CH, _4_-CH, _3_-CH)), 1.348 (m, 2H, 2 _3_-CH), 1.107 (s, 6H, 2 _9_-CH_3_), 0.934 (s, 6H, 2 _8_-CH), 0.788 (d, J = 10 Hz, 2H, 2 _7_-CH); ^13^C-NMR (CDCl_3_, 100 Hz) δc: 62.82 (2C-_11_), 51.79 (2C-_α_), 46.26 (2C-_2_), 41.01 (2C-_5_), 38.56 (2C-_6_), 38.25 (2С-_1_), 33.31 (2C-_10_), 30.14 (2C-_7_), 27.91 (2C-_9_), 26.05 (2C-_4_), 23.36 (2C-_8_), 22.15 (2C-_3_); IR (KBr) *v* (cm^−1^) 2996.90, 2984.47, 2929.30, 2888.99, 2865.64, 1469.01, 1448.93, 1362.64; LC-MS, C24H44NI, m/z 346.34(M^+^-I).

Spectroscopic analysis of compound **4c**: bis-hydronopyl piperidine ammonium iodide, white solid, m.p. 252.1–258.9 °C; Yield, 84%.^1^H-NMR (CDCl3, 400 MHz) δH: 3.712 (m. 4H, 2 _α_-CH_2_), 3.381 (m, 4H, 2 _11_-CH_2_), 2.321 (m, 2H, 2 _2_-CH), 2.061~1.694 (m, 22H, 2(_7_- CH, _10_-CH_2_, _5_-CH, _1_-CH, _4_-CH_2_, _3_-CH), 2 _β_-CH_2_, _γ_-CH_2_), 1.418 (m, 2H, 2 _3_-CH), 1.181 (s, 6H, 2 _9_-CH_3_), 0.999 (s, 6H, 2 _8_-CH_3_), 0.878 (d, J = 9.6 Hz, 2H, 2 _7_-CH); ^13^C-NMR (CDCl_3_, 100 Hz) δc: 59.20 (2C-_11_), 57.31 (2C-_α_), 46.46 ( 2C-_2_), 40.99 (2C-_5_), 38.55 (2C-_6_), 38.44 (2C-_1_), 33.33 (2C-_10_), 29.11 (2C-_7_), 27.90 (2C-_9_), 26.01 (2C-_4_), 23.35 (2C-_8_), 22.25 (2C-_3_), 20.45 (C-_γ_), 19.98 (2C-_β_); IR (KBr) *v* (cm^−1^) 2978.50, 2936.90, 2899.96, 2865.87, 1468.43, 1382.56, 1366.37; LC-MS, C27H48NI, *m/z* 386.37 (M^+^-I). 

Spectroscopic analysis of compound **4d**: bis-hydronopyl morpholine ammonium chloride, white solid, m.p. 240.2–246.9 °C; Yield, 81%.^1^H-NMR (CDCl_3_, 400 MHz) δH: 4.312 (t, J = 12 Hz, 4H, 2 _β_-CH_2_), 3.958 (d, J = 10.8 Hz, 4H, 2 _11_ = CH_2_), 3.411 (d, J = 11.6 Hz, 4H, 2 _α_-CH_2_), 2.339 (m, 2H, 2 _2_-CH), 1.992~1.746 (m, 16H, 2(_7_-CH, _10_-CH_2_, _5_-CH, _1_-CH, _4_-CH_2_, _3_-CH)), 1.464 (m, 2H, 2 _3_-CH), 1.178 (s, 6H, 2 _9_-CH_3_), 1.028 (s, 6H, 2 _8_-CH_3_), 0.868 (d, J = 9.6 Hz, 2H, 2 _7_-CH); ^13^C-NMR (CDCl_3_, 100 Hz) δc: 63.10 (2C-_β_), 56.55 (2C-_11_), 51.25 (2C-_α_), 45.46 (2C-_2_), 40.71 (2C-_5_), 38.44 (2C-_1_), 38.17 (2C-_6_), 32.86 (2C-_10_), 29.97 (2C-_7_), 27.48 (2C-_9_), 25.63 (2C-_4_), 22.72 (2C-_8_), 21.51 (2C-_3_); IR (KBr) *v* (cm^−1^) 2974.57, 2909.85, 2867.29, 2530.74, 2439.84, 1452.12, 1406.70, 1406.70, 1126.11, 1090.08; LC-MS, C26H46NOCl, *m/z* 388.35 (M^+^-Cl).

Spectroscopic analysis of compound **4e**: bis-hydronopyl morpholine ammonium bromide, white solid, m.p. 246.8–250.7 °C; Yield, 80%. ^1^H-NMR (CDCl_3_, 400 MHz) δH: 4.357 (t, J = 11.2 Hz, 4H, 2 _β_-CH_2_), 3.975 (d, J = 12.8 Hz, 4H, 2 _11_-CH_2_), 3.452 (d, J = 11.6 Hz, 4H, 2 α-CH_2_), 2.334 (m, 2H, 2 _2_-CH), 2.006~1.807 (m, 16H, 2 (_7_-CH_2_, _10_-CH_2_, _5_-CH_2_, _1_-CH_2_, _4_-CH_2_, _3_-CH)), 1.469 (m, 2H, 2 _3_-CH), 1.163 (s, 6H, 2 _9_-CH_3_), 1.015 (s, 6H, 2 _8_-CH_3_), 0.849 (d, J = 9.6 Hz, 2H, 2 _7_-CH), ^13^C-NMR (CDCl_3_, 100 Hz) δc: 62.96 (2C-_β_), 56.53 (2C-_11_), 51.19 (2C-_α_), 45.44 (2C-_2_), 40.69 (2C-_5_), 38.36 (2C-_1_), 38.16 (2C-_6_), 32.85 (2C-_10_), 29.93 (2C-_7_), 27.46 (2C-_9_), 25.61 (2C-_4_), 22.78 (2C-_8_), 21.46 (2C-_3_); IR (KBr) *v* (cm^−1^) 2973.91, 2909.09, 2641.53, 2593.82, 2558.96, 2470.55, 1468.60, 1442.80, 1126.62, 1090.98; LC-MS, C26H46NOBr *m/z* 388.35 (M^+^-Br). 

Spectroscopic analysis of compound **4f**: bis-hydronopyl morpholine ammonium iodide white solid, m.p. 265.4–271.6 °C; Yield, 82%. ^1^H-NMR (CDCl3, 400 MHz) δH: 4.049 (s; 4H, 2 β-CH_2_), 3.712 (s, 4H, 2 α-CH2), 3.602~ 3.474 (m, 4H, 2 11-CH_2_). 2.305 (m, 2H, 2-CH), 2.084~1.739 (m, 16H, 2(7-CH, 10-CH, 5-CH, 1-CH, 4-CH, 3-CH)), 1.393 (m, 2H, 2 3-CH), 1.158 (s, 6H, 2 9-CH_3_), 0.988 (s, 6H, 2 8-CH_3_), 0.861 (d, J = 10 Hz, 2H, 2 _7_-CH). ^13^C-NMR (CDCl_3_, 100 Hz) δc: 60.09 (2C-β), 58.07 (2C-α), 57.94 (2C-11), 45.88 (2C-2), 40.49 (2C-5), 38.04 (2C-6), 37.83 (2C-10), 32.82 (2C-10), 28.67 (2C-7), 27.39 (2C-9), 25.53 (2C-4), 22.90 (2C-8), 21.69 (2C-3). IR (KBr) *v* (cm^−1^) 2981.81, 2936.92, 2904.26, 2869.57, 1128.74, 906.29; LC-MS, C26H46NOI: *m/z* 388.35 (M^+^-I).

### 4.3. Antifungal Activity

The antifungal activity of the synthetic compounds was evaluated in vitro against the ten phytopathogenic fungi using a mycelia growth inhibition method [38]. All the *β*-PQA salts were dissolved in DMSO (dimethyl sulfoxide) and then added to PDA medium that was prepared and sterilized to obtain a series of concentrations (200, 100, 50, 25, 12.5, 6.25 and 3.125 μg/mL). The mycelial disk (5 mm) of phytopathogenic fungi was inoculated on PDA plates and then incubated at 25 °C in the dark. Each sample was measured in triplicate and its diameters (mm) of inhibition zones were measured by the cross-bracketing method. DMSO (0.4%, *v*/*v*) served as a blank control and the fungicide chlorothalonil was used as a positive control. The inhibition rate was calculated according to the following equation:Inhibition rate (%) = [(C − T)/(C − 5 mm)] × 100(1)
where C and T represents the average diameter of the fungus cake (mm) of the control and treatment groups, respectively.

The EC_50_ value was defined as the concentration required for 50% inhibition of mycelial growth. The regression equation was obtained by using the natural logarithm of the compound concentrations as the abscissa, and the biometric probability value calculated by inhibition rate as the ordinate. The EC_50_ value was calculated using the regression equation when the inhibition rate value reached 50%.

### 4.4. Assay of Minimal Inhibitory Concentration

The viability of bacterial cells when exposed to varying concentrations of *β*-PQA salts was analyzed in a 96-well plate using the resazurin assays to evaluate MIC [39,40]. A stock solution (2.56 mg/mL) of the tested compounds was prepared by dissolving them into the saline solution, and then serially diluted into a series of concentrations (two-fold dilution) with nutrient broth. 100 μL of nutrient broth, 50 μL of bacterial suspension (1.0 × 10^8^ CFU/mL, D_625nm_ = 0.08–0.10), 50 μL of the compound and 50 μL of a 0.1% resazurin solution were successively added to each well and mixed by vortex shaking. After that, the mixtures were kept in an incubator for 24 h at a temperature of 37 °C. Saline solution (0.9%) served as a blank control and the standard drug chloramphenicol was used as a positive control.

### 4.5. Anticancer Activity

The human cancer cell lines, MCF-7 (breast cancer cells) and HCT-116 (colon cancer cell lines), used for the study were procured from the BeNa Culture Collection (Beijing, China) and iCell Bioscience Inc. (Shanghai, China), respectively. The MCF-7 and HCT-116 cells lines were cultured with the DMEM and McCOY’S5A medium containing 10% of fetal bovine serum (FBS) with 5% CO_2_ in a 37 °C incubator (WCI-180, Wiggens, Germany), respectively. The Cell Counting Kit 8 (CCK-8) assay was used to assess the effect of the compounds on cell viability according to the method in [41,42] with some modifications. Briefly, 4 × 10^3^ cells/well were exposed to different concentrations of sorafenib (positive control) and *β*-PQA salts (200, 100, 50, 25 and 12.5 μM) in the 96-well plates for 24 h. Then, the medium were removed, and PBS was added to wash each well. Cells were incubated at 37 °C for 2 h with 100 μl of 10% CCK-8 solution per well (Invigentech IV08-100, California, USA). The OD values were recorded at 450 nm with a Microplate reader (TECAN SPARK 10M, Männedorf, Switzerland). Data were expressed as the percentage of viable cells in treated relative to nontreated conditions. 

### 4.6. Determination of Cell Membrane Permeability

The mycelia and strains were prepared as follows: the *F. oxysporum f.sp. niveum* mycelial disks (5 mm) were placed in 50 mL PD (potato dextrose) liquid medium at 28 °C for 3 days with 150 rpm shaking. The mycelium of *F. oxysporum f.sp. niveum* were obtained by suction filtration. The strains of *P. aeruginosa* (200 μL) were placed in 50 mL of LB liquid medium at 37 °C for 24 h with 150 rpm shaking, and the strains were obtained by centrifugation.

The influence of compound **4a** on the cell membrane relative permeability rate of *F. oxysporum f.sp. niveum* and *P. aeruginosa* was evaluated according to the previous research [43]. The mycelia and strains were treated with EC_50_ (4.5 μg/mL) and MIC (0.625 μg/mL) of compound **4a**, which were followed by the measurement of conductivities at 0 (E_0_), 20, 40, 60, 90, 120 and 180 min (E_1_), and the conductivity (E_2_) was measured after boiling and subsequent cooling, respectively. The relative permeability rate of the cell membrane was calculated by the following equation:Relative electric conductivity (%) = [(E_1_ − E_0_)/(E_2_ − E_0_)] × 100(2)

### 4.7. Scanning Electron Microscope (SEM) Observations

SEM observations were conducted according to the method of previous work with minor modifications [29,44]. Briefly, 5 mm diameter mycelial disks of *F. oxysporum f.sp. niveum* were cut from the periphery of the colony grown on PDA containing an EC_50_ of 4.5 μg/mL of compound **4a** and 0.5% DMSO (blank control), respectively. *P. aeruginosa* was cultured (10^7^ CFU/mL) with compound **4a** (MIC: 0.625 μg/mL) or 0.5% DMSO (blank control). Cells were harvested after incubation for 12 h at 37 °C. The samples were fixed in 2.5% (*w*/*v*) glutaraldehyde at 4 °C for 12 h and washed twice with PBS, and then the samples were dehydrated in a graded series of ethanol (30%, 50%, 70% and 95% respectively, 10 min for each alcohol dilution). After drying at critical point and gold coating, the samples were examined by SEM (ZEISS, Sigma300, Jena, Germany).

### 4.8. Detection of Intracellular ATP

The *F. oxysporum f.sp. niveum* mycelia and *P. aeruginosa* strains’ preparation were conducted using the same procedure as described for the cell membrane permeability experiment. The mycelia and strains were incubated with sample at the EC_50_ of 4.5 μg/mL (at 28 °C) and MIC of 0.625 μg/mL (at 37 °C) for 6 h, respectively. The content of intracellular ATP was measured by A016-1-1 assay kits according to the manufacturer’s instructions of Nanjing Jiancheng Bioengineering Institute (Nanjing, Cheng). 

## 5. Conclusions

In summary, six *β*-PQA salts were successfully synthesized by a convenient and efficient procedure. The bioassay results revealed that most *β*-PQA salts had high antifungal, antibacterial and anticancer activities. Compounds **4a** and **4b** demonstrated excellent and broad-spectrum antifungal and antibacterial activities. In addition, compounds **4a**, **4b**, **4c** and **4f** exhibited remarkable anticancer activity against HCT-116 and MCF-7 cell lines. This study demonstrated that the introduction of methyl groups or a piperidine group on the hydrophilic group N^+^ of the *β*-PQA salts played a key role in the improvement of the antifungal, antibacterial and anticancer activities. Moreover, the compound **4a** could destroy the cell membrane and affect the ATP activity, which could partly explain its antimicrobial mechanism. All results indicate that compound **4a** is a promising compound for the development of a new generation of antifungal, antibacterial and anticancer agents, and in vivo antimicrobial tests and potential adverse effects on human health need to be further explored.

## Figures and Tables

**Figure 1 ijms-22-11299-f001:**
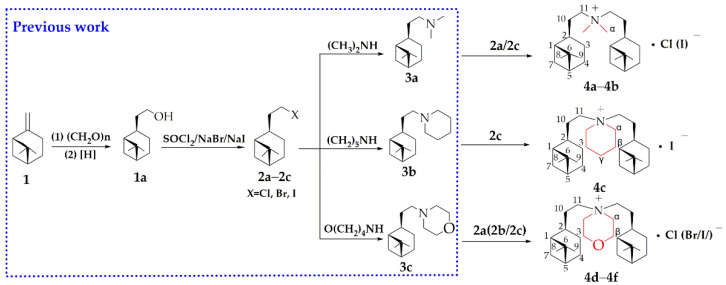
Synthetic routes of *β*-pinene quaternary ammonium salts (**4a**–**4f**).

**Figure 2 ijms-22-11299-f002:**
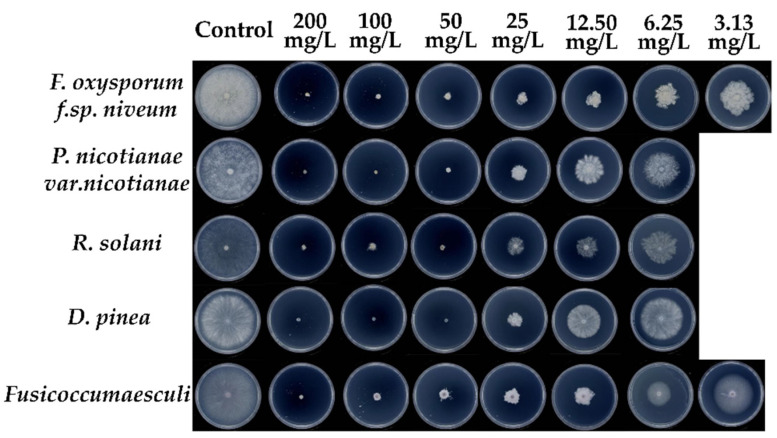
In vitro antifungal activities of compound **4a** against five fungi.

**Figure 3 ijms-22-11299-f003:**
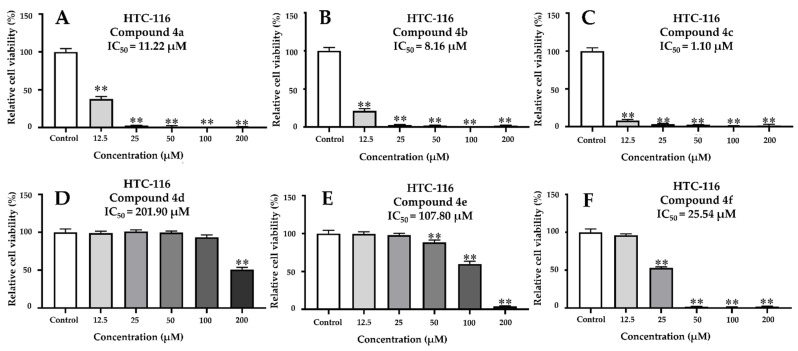
Dose-dependent effects of ascending amounts of *β*-PQA salts against HTC116 cancer cell lines on the percentage inhibition of cell proliferation. (**A**) HTC-116 cells were treated with compound **4a** for 24 h. (**B**) HTC-116 cells were treated with compound **4b** for 24 h. (**C**) HTC-116 cells were treated with compound **4c** for 24 h. (**D**) HTC-116 cells were treated with compound **4d** for 24 h. (**E**) HTC-116 cells were treated with compound **4e** for 24 h. (**F**) HTC-116 cells were treated with compound **4f** for 24 h. Dunnett’s multiple comparisons test was used to analyze the difference between the treatment group and the control group, (**) *p* < 0.01.

**Figure 4 ijms-22-11299-f004:**
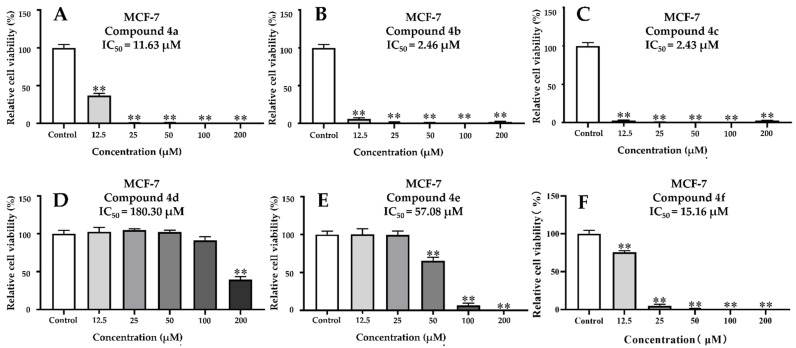
Dose-dependent effects of ascending amounts of *β*-PQA salts against MCF-7 cancer cell lines on the percentage inhibition of cell proliferation. (**A**) MCF-7 cells were treated with compound **4a** for 24 h. (**B**) MCF-7 cells were treated with compound **4b** for 24 h. (**C**) MCF-7 cells were treated with compound **4c** for 24 h. (**D**) MCF-7 cells were treated with compound **4d** for 24 h. (**E**) MCF-7 cells were treated with compound **4e** for 24 h. (**F**) MCF-7 cells were treated with compound **4f** for 24 h. Dunnett’s multiple comparisons test was used to analyze the difference between the treatment group and the control group, (**) *p* < 0.01.

**Figure 5 ijms-22-11299-f005:**
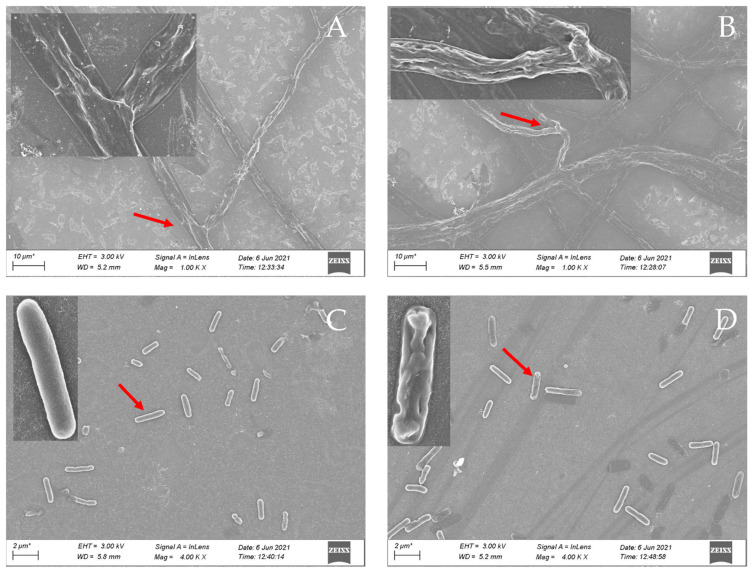
SEM of the form of *F. oxysporum f.sp. niveum* and *P. aeruginosa*. The forms of *F. oxysporum f.sp. niveum* were (**A**) untreated control and (**B**) treated with compound **4a** at EC_50_ × 1000, and the upper left multiple was ×4000. The forms of *P. aeruginosa* were (**C**) untreated control and (**D**) treated with compound **4a** at MIC ×4000, and the upper left multiple was ×20,000.

**Figure 6 ijms-22-11299-f006:**
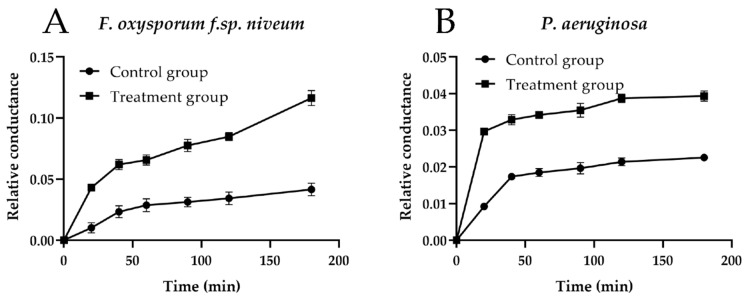
Relative electric conductivity of compound **4a** against *F. oxysporum f.sp. niveum* (**A**) and *P. aeruginosa* (**B**).

**Figure 7 ijms-22-11299-f007:**
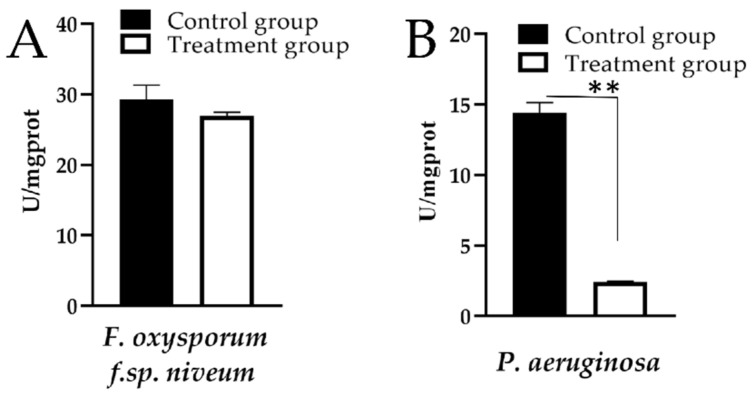
Effect of compound **4a** on ATPase activity against *F. oxysporum f.sp. niveum* (**A**) and *P. aeruginosa* (**B**). Statistical analysis: Student’s T test, (**) *p* < 0.01.

**Table 1 ijms-22-11299-t001:** The 50% inhibition of mycelial growth (EC_50_) for *β*-pinene quaternary ammonium salts against ten fungi.

Fungi	The Value of EC_50_ (μg/mL)						
	4a	4b	4c	4d	4e	4f	Ch.
*F. oxysporum f.sp. niveum*	4.5	17.42	40.82	49.63	50.53	391.04	3.98
*P. nicotianae var.nicotianae*	10.92	15.1	10.54	229.37	94.06	189.69	1.81
*C. acutatum*	27.09	18.21	19.58	52.94	33.31	168.86	27.09
*R. solani*	9.45	10.97	9.36	24.62	11.77	74.83	0.43
*C. versicolor*	31.98	27.83	77.68	113.18	296.61	26411.41	2.41
*F. verticillioides*	15.09	10.78	20.67	46.15	36.81	61.94	0.67
*D. pinea*	10.82	18.85	14.32	906.84	81.94	203.9	1.74
*P. vaporaria*	18.01	13.71	11.02	181.88	165.09	306.31	0.07
*Fusicoccumaesculi*	6.34	5.39	10.25	32.56	40.37	93.84	0.09
*C. gloeosprioides*	33.76	18.98	21.13	32.12	36.03	398.38	2.05

Note: *Fusarium oxysporum f*.*sp*. *niveum*: *F*. *oxysporum f*.*sp*. *niveum*; *Phytophthora nicotianae var*.*nicotianae*: *P*. *nicotianae var*.*nicotianae*; *Colletotrichum acutatum*: *C*. *acutatum*; *Rhizoctonia solani*: *R*. *solani*; *Coriolus versicolor*: *C*. *versicolor*; *Fusarium verticillioides*: *F*. *verticillioides*; *Diplodia pinea*: *D*. *pinea*; *Poria vaporaria*: *P*. *vaporaria*; *Colletotrichum gloeosprioides*: *C*. *gloeosprioides*; *Ch*.: *Chlorothalonil*.

**Table 2 ijms-22-11299-t002:** The minimum inhibitory concentration (MIC) for *β*-pinene quaternary ammonium salts against four bacteria.

Bacteria	The Value of MIC (μg/mL)						
	4a	4b	4c	4d	4e	4f	Chloramphenicol
*E. coli*	2.5	20	80	160	160	320	5
*P. aeruginosa*	0.625	80	160	160	80	320	80
*S. aureus*	1.25	80	80	320	80	320	0.625
*B. subtilis*	1.25	5	80	160	160	640	0.625

Note: *E*. *coli*, *Escherichia coli*; *P*. *aeruginosa*, *Pseudomonas aeruginosa*; *S*. *aureus*, *Staphylococcus aureus*; *B. subtilis*, *Bacillus subtilis.*

**Table 3 ijms-22-11299-t003:** IC_50_ values of the tested compounds against HCT-116 and MCF-7 cell lines.

Compound	IC_50_ (μM)	
	HCT-116	MCF-7
**4a**	11.22	11.63
**4b**	8.16	2.46
**4c**	1.10	2.43
**4d**	201.90	180.30
**4e**	107.80	57.08
**4f**	25.54	15.16
Sorafenib	20.79	13.92

Note: Cells were treated with the test compounds or vehicle for 24 h. Two human cancer cell lines were used: HCT-116 (human colon cancer cell lines) and MCF-7 (human breast cancer cells). Sorafenib was used as a positive control in the anticancer screening assay.

## Data Availability

The authors declare that (the/all other) data supporting the findings of this study are available within the article (and its Appendix A).

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
