# Peer review of "Design, Synthesis, Antibacterial, Antifungal and Anticancer Evaluations of Novel β-Pinene Quaternary Ammonium Salts"

_ijms, 2021, doi:10.3390/ijms222011299_

Round 1

Reviewer 1 Report

The manuscript “Design, Synthesis, Antibacterial, Antifungal and anticancer evaluations of Novel β-pinene quaternary ammonium salts” by Zhang et al. reports interesting results on the antifungal, antibacterial and anticancer potential of newly synthesized β-pinene quaternary ammonium salts. The authors found that β-PQA salts they synthesized possess high antifungal, antimicrobial and tumour cell antiproliferative activities. For compound 4a (bis-hydronopyl dimethyl ammonium chloride) they showed that this compound causes mycelium abnormalities, increases the permeability of cell membranes and inhibits the activity of ATP in F. oxysporum f.sp. niveum and P.aeruginosa.

Overall the manuscript is well written and can be accepted after some minor corrections.

Minor Comments and Suggestions:

Lane 101: Should read “…compound 4a, 4b and 4c exhibited broad-spectrum antifungal …”

Lanes 129-130: Should read “…following order: 4a> 4b > 4c > 4e > 4d > 4f.”

Lanes 217-218: Should read “Interestingly, as compared with control group, the intracellular ATP activity of the strains treated with compound 4a was decreased (Figure 7).”

Lanes 268-272: The authors might consider to rewrite this part as following: “We found that the activity of ATP of F. oxysporum f.sp. niveum and P.aeruginosa treated with compound 4a decreased, indicating that compound 4a can disrupt their energy metabolism. So the energy metabolism disruption of microbial is probably one of the main antimicrobial mechanism of β-PQA salts, especially for the compound 4a."

Author Response

Response to Reviewer 1 Comments

The manuscript “Design, Synthesis, Antibacterial, Antifungal and anticancer evaluations of Novel β-pinene quaternary ammonium salts” by Zhang et al. reports interesting results on the antifungal, antibacterial and anticancer potential of newly synthesized β-pinene quaternary ammonium salts. The authors found that β-PQA salts they synthesized possess high antifungal, antimicrobial and tumour cell antiproliferative activities. For compound 4a (bis-hydronopyl dimethyl ammonium chloride) they showed that this compound causes mycelium abnormalities, increases the permeability of cell membranes and inhibits the activity of ATP in F. oxysporum f.sp. niveum and P.aeruginosa.

Overall the manuscript is well written and can be accepted after some minor corrections.

Response: Thanks for your affirmation on the manuscript. We have revised the comments and suggestions and also the full manuscript carefully. Thanks again for your kind suggestions.

Minor Comments and Suggestions:

Lane 101: Should read “…compound 4a, 4b and 4c exhibited broad-spectrum antifungal …”

Response: Thanks. We have done it.

Lanes 129-130: Should read “…following order: 4a> 4b > 4c > 4e > 4d > 4f.”

Response: Thanks. We have done it.

Lanes 217-218: Should read “Interestingly, as compared with control group, the intracellular ATP activity of the strains treated with compound 4a was decreased (Figure 7).”

Response: Thanks. We have done it.

Lanes 268-272: The authors might consider to rewrite this part as following: “We found that the activity of ATP of F. oxysporum f.sp. niveum and P.aeruginosa treated with compound 4a decreased, indicating that compound 4a can disrupt their energy metabolism. So the energy metabolism disruption of microbial is probably one of the main antimicrobial mechanism of β-PQA salts, especially for the compound 4a.

Response: Thanks for your constructive suggestion. We have done it accordingly.

Reviewer 2 Report

The results and discussions part should be a little more detailed.

How can the information presented in lines 111-115 or 138-141 be explained on the basis of a structure-biological activity relationship? Why the presence of halide ions (Br- > Cl-) and groups (two methyl groups, piperidine group) into the hydrophilic group N+ of the β-PQA salts was conducive to improve the antibacterial or antifungal activity?

Regarding antitumor activity, studies should be supplemented with testing for cytotoxic activity in normal human cells to monitor the toxicity of these compounds in this type of cell as well.

Author Response

Response to Reviewer 2 Comments

Comments and Suggestions for Authors

The results and discussions part should be a little more detailed.

Response: Thanks. We have done it accordingly.

How can the information presented in lines 111-115 or 138-141 be explained on the basis of a structure-biological activity relationship? Why the presence of halide ions (Br- > Cl-) and groups (two methyl groups, piperidine group) into the hydrophilic group N+ of the β-PQA salts was conducive to improve the antibacterial or antifungal activity?

Response: Thanks for your constructive suggestion. To reduce the doubts about it and make the manuscript more readable, we have added some detailed comments and some references in the discussion (Seen in the paragraph two).

Regarding antitumor activity, studies should be supplemented with testing for cytotoxic activity in normal human cells to monitor the toxicity of these compounds in this type of cell as well.

Response: Thanks for this comment. We also think that tests for cytotoxic activity in normal human cells is necessary. The research focus of this article is to provide theoretical basis for developing new antibacterial agents, and all the results show that these compounds have the potential to become a new type of antibacterial or antifungal agent, we just explore their anticancer activity preliminarily in this study. Considering the significant effects of these compounds on anticancer, we will further conduct related experiments to explore the anticancer mechanism of these compounds, including the study of the toxicity to normal human cells.
